# Shoulder girdle formation and positioning during embryonic and early fetal human development

**Sayaka Tanaka[1], Rino Sakamoto[1], Toru Kanahashi[1], Shigehito Yamada[1,2], Hirohiko Imai[3], Akio Yoneyama[4], Tetsuya Takakuwa[1] \***

1 Human Health Science, Graduate School of Medicine, Kyoto University, Kyoto, Japan, 2 Congenital Anomaly Research Center, Graduate School of Medicine, Kyoto University, Kyoto, Japan, 3 Department of Systems Science, Graduate School of Informatics, Kyoto University, Kyoto, Japan, 4 SAGA Light Source, Saga, Japan

\* tez@hs.med.kyoto-u.ac.jp

**Data Availability Statement:** All relevant data are within the manuscript and its Supporting Information files.

## Abstract

Positional information on the shoulder girdle (the clavicle and scapula) is important for a better understanding of the function of the upper limb in the locomotive system as well as its associated disease pathogenesis. However, such data are limited except for information on the axial position of the scapula. Here, we describe a three-dimensional reconstruction of the shoulder girdle including the clavicle and scapula, and its relationship to different landmarks in the body. Thirty-six human fetal specimens (crown-rump length range: 7.6–225 mm) from the Kyoto Collection were used for this study. The morphogenesis and three-dimensional position of the shoulder girdle were analyzed with phase-contrast X-ray computed tomography and magnetic resonance imaging. We first detected the scapula body along with the coracoid and humeral head at Carnegie stage 18; however, the connection between the body and coracoid was not confirmed at this stage. During development, all landmarks on the shoulder girdle remained at the same axial position except for the inferior angle, which implies that the scapula enlarged in the caudal direction and reached the adult axial position in the fetal period. The scapula body was rotated internally and in the upward direction at the initiation of morphogenesis, but in the fetal period the scapula body was different than that in the adult position. The shoulder girdle was located at the ventral side of the vertebrae at the time of initial morphogenesis, but changed its position to the lateral side of the vertebrae in the late embryonic and fetal periods. Such a unique position of the shoulder girdle may contribute to the stage-specific posture of the upper limb. Adequate internal and upward rotation of the scapula could help in reducing the shoulder width, thereby facilitating childbirth. The data presented in this study can be used as normal morphometric references for shoulder girdle evaluations in the embryonic and fetal periods.

**Funding:** JP26220004, JP16K15535, JP17H05294, JP18K07876 from the Japan Society for the Promotion of Science for Takakuwa T.

**Competing interests:** The authors have declared that no competing interests exist.

## Introduction

The shoulder girdle (pectoral girdle) is the set of bones in the appendicular skeleton, which anchors the upper limb on each side to the axial skeleton [1, 2]. In humans, it consists of the clavicle and scapula. The sternoclavicular joint (SCJ) links the sternum with the clavicle at the anterior part of the chest and is the only true anatomical joint between the shoulder girdle and axial skeleton. The acromioclavicular joint (ACJ) links the clavicle with the acromion on the scapula, while the shoulder or the glenohumeral joint provides articulation between the glenoid cavity of the scapula and the appendicular skeleton (the humerus). Movements in these joints can change the position of each component. More than 10 muscles are attached to the scapula, and collectively, they provide stability to the scapula as well as complex movements, such as elevation, depression, retraction (adduction), protraction (abduction), upward rotation, and downward rotation [1–4]. Due to such movability, it is difficult to describe a default position of the shoulder girdle. In adults, the triangular scapula body can be seen from the dorsal side while the clavicle can be seen from the ventral side, located obliquely at the upper border of the chest. The scapula is located at the axial position between the Th2 and Th7 vertebrae. The medial border is almost parallel to the vertebrae column (the cranial-caudal axis). From the cranial view, the angle of the scapular plane, the clavicle orientation, is around 30 degrees. Thus, the angle between the scapular plane and clavicle from the cranial view is around 60 degrees.

Several studies have described the morphogenesis of the human shoulder girdle. Lewis (1902) [5] precisely described the morphogenesis of the upper limb in the embryonic period, including the skeleton, muscles, and nerves. The scapula and clavicle were illustrated as skeletal components. Regarding the scapula, the author paid particular attention to the cranial position of the scapula in the initial blastemal and chondrogenous phases [5]. Blechschmidt, Müller and O'Rahilly, and O'Rahilly et al. also noted the axial position of the scapula [6–9]. Recent studies have primarily described the morphogenesis of the shoulder joint (between the scapula and humerus) [10]. Moreover, specific structures such as the glenoid labrum, the long head (caput longum) of the biceps tendon [11] and the rotator interval [12] have also been described in detail.

Morphogenesis of the shoulder girdle has been described in the context of the upper limb development in most human embryology books/documents. However, both the scapula and clavicle have different embryonic origins, and their development is controlled by a genetic regulation that is different than that of the upper limb [13]. For instance, the scapular origin and genetic control are more similar to those of the spine, which could explain the high concordance in vertebral and scapular anomalies, particularly, in those anomalies that are distant from the immediate vicinity of the scapula such as diastematomyelia and lumbosacral spina bifida occulta [14, 15]. Therefore, the shoulder girdle should be described and analyzed separately from other parts of the upper limb.

Currently, there is limited information on the 3-D morphogenesis and the position of the whole shoulder girdle except for information on the axial position of the scapula in the embryonic and fetal periods [5]. Scapular dyskinesis and scapular deformity related to obstetric brachial palsy are known as the deviation of the scapula from the "normal" position [16, 17]. In addition, the shoulder girdle morphology (size) and position affect childbirth and neonatal development. Neonates with shoulder dystocia have a significantly greater shoulder-to-head ratio than neonates with normal development [18]. The clavicle fracture and brachial palsy are known complications of childbirth [19]. Since neonatal positioning including the shoulder girdle affects vital functions such as heart and respiratory rates [20], congenital anomalies related to the malformation and abnormal position of the scapula are frequently observed [13].

Collectively, these observations indicate that elucidating the shoulder girdle morphogenesis and position in the fetal period is important for a better understanding of the function of the upper limb in the locomotive system and its associated disease pathogenesis. In this study, therefore, we described a 3-D reconstruction of the shoulder girdle formation and positioning including the clavicle and scapula and its relationship to different landmarks in the body.

## Materials and methods

### Human fetal specimens

A total of 36 human embryo and fetal specimens (23 embryo specimens from Carnegie stage (CS) 16 to CS23 [crown-rump length (CRL) range: 7.6–28.0 mm] and 13 fetal specimens [CRL range: 29.8–225 mm]) from the Kyoto Collection at the Congenital Anomaly Research Center of Kyoto University, Japan [21–23] were used for this study (Table 1). These specimens were measured, examined, and staged according to the criteria proposed by O'Rahilly and Müller [24].

Most specimens stored at the Kyoto Collection were acquired when pregnancy was terminated for socioeconomic reasons and in accordance with the Maternity Protection Law of Japan. Samples were collected between 1963 and 1995 in accordance with relevant regulations of those time periods. For instance, written informed consent was not required from parents at that time. Instead, parents provided verbal informed consent to have these specimens deposited, and each participant's consent was documented in the medical record. All samples were anonymized and de-identified. The ethics committee of the Kyoto University Faculty and Graduate School of Medicine approved this study, which used human embryo and fetal specimens (E986, R0316).

### Image acquisition

The image acquisition parameters for three-dimensional (3-D) phase-contrast X-ray—computed tomography (PCX-CT) had been described previously [25]. Briefly, specimens were visualized using a phase-contrast imaging system fitted with a crystal X-ray interferometer [26]. The system was installed in a vertical wiggler beamline (PF BL-14C) at a Photon Factory in Tsukuba, Japan. The white synchrotron radiation emitted from the vertical wiggler was

**Table 1. Specimens analyzed in this study.**

| Sample (stage) | n | CRL (mm) | PCX-CT/MRI |
|---|---|---|---|
| Embryonic period | 23 | | |
| CS16 | 3 | 7.6~9.2 | PCX-CT |
| CS17 | 3 | 10.3~11.9 | PCX-CT |
| CS18 | 3 | 14.3~18.0 | PCX-CT |
| CS19 | 3 | 14.1~15.5 | PCX-CT |
| CS20 | 3 | 17.0~19.3 | PCX-CT |
| CS21 | 2 | 18.8~23.0 | PCX-CT |
| CS22 | 3 | 22.0~24.5 | PCX-CT(1)/MRI(2)* |
| CS23 | 3 | 23.4~28.0 | PCX-CT(2)/MRI(1)* |
| Fetal period | 13 | 29.8~225 | MRI |

CS; Carnegie stage, CRL; Crown-rump length, PCX-CT: phase-contrast X-ray—computed tomography, MRI: Magnetic resonance images.

* number in parenthesis indicates the sample number.

monochromated by a Si (220) double-crystal monochromator, horizontally magnified by an asymmetric crystal, and introduced to the imaging system. The interference patterns were detected by large-area X-ray imagers that consisted of a 30-μm scintillator, a relay lens system, and a water-cooled charge-coupled device camera PixelVision 2Kx2K CCD camera (PixelVision, Hasselt, Belgium) [36 × 36 mm field of view, 2048 × 2048 pixels, 18 × 18 μm each] [27], as well as a 100-μm CsI scintillator, optical fiber, and Zyla sCMOS HF (Oxford Instruments, Abingdon, Oxfordshire, England) [16 × 13 mm field of view, 2560 × 2160 pixels, 6.5 × 6.5 μm each]. The X-ray energy was tuned at 17.8 keV, and an exposure time of 3–5 s was used to obtain one interference pattern. The average intensity was approximately 5000 counts/pixel, which allowed for high-resolution observations within a reasonable measurement time period.

Magnetic resonance (MR) images were acquired using a 7-T MR system (BioSpec 70/20 USR; Bruker BioSpin MRI GmbH, Ettlingen, Germany) and 3-T MR system (MAGNETOM Prisma; Siemens Healthineers, Erlangen, Germany). The 7-T MR system was equipped with $^1$H quadrature transmit-receive volume coils of 35 and 72 mm diameters (T9988 and T9562; Bruker BioSpin MRI GmbH, Ettlingen, Germany) [28]. The 3-D T1-weighted images were acquired using a fast low-angle shot pulse sequence with the following parameters: repetition time, 30 ms; echo time, 4.037–6.177 ms; flip angle, 40˚; the field of view, 22.5 × 15.0 × 15.0 to 42.0 × 28.0 × 28.0 μm$^3$; matrix size, 636 × 424 × 424 to 768 × 512 × 512; and isotropic spatial resolution, 35.4–54.7 μm$^3$.

PCX-CT was used to acquire the 3-D images of specimens between CS16 and CS23, whereas MRI was used for imaging specimens at CS23 and later stages. The 7-T MR system was used to acquire the 3-D images of fetal specimens with CRL ranging between 29.8 and 112 mm. In contrast, the 3-T MR system was used for imaging fetal specimens with CRL > 116 mm. The image acquisition method was selected based on the desired resolution and the volume of the specimen. For instance, PCX-CT was used to acquire images at a higher resolution than what could be obtained using MRI. However, PCX-CT could not be used to acquire images of specimens with large volumes. Thus, CS22 and CS23 samples represented the upper limit with respect to the size of specimens that could be examined with PCX-CT.

## Image analysis, anatomical landmarks, and position of the upper girdle

PCX-CT and MRI data from selected specimens were precisely analyzed using serial 2-D images and reconstructed 3-D images. The 3-D images of the scapula and clavicle were manually reconstructed using the Amira software, version 5.5.0 (Visage Imaging GmbH, Berlin, Germany). The 3-D coordinates were initially assigned by examining the voxel position on 3-D images.

The 3-D coordinates were acquired for both the scapula and clavicle. The ventral tip of the first rib, and the most dorsal point on the vertebral body midline between the fourth cervical and 12th thoracic vertebrae (C4-Th12) were used as an internal reference.

The vector between the first and fifth thoracic vertebrae (Th1 and Th5, respectively) was defined as the cranial-caudal axis (Z-axis) (Table 2). The normal vector of the z-axis, which runs between the midpoint of the tip of the first rib pairs, was defined as the dorso-ventral axis (y-axis). The transverse axis (x-axis) was calculated as the outer product of the z and y axes. Th1 was defined as the origin.

The following reference points were selected to acquire 3-D coordinates; ACJ: Acromio-clavicular joint, glc: Glenoid cavity, ifa: Inferior angle, it: Infraglenoid tubercle, mss: medial end the scapula spine, SCJ: Sterno-clavicular joint, spa: Superior angle, and st: supraglenoid tubercle. The glenoid cavity was calculated as the midpoint between the supraglenoid tubercle and Infraglenoid tubercle.

**Table 2. Aaronyms and abbreviations for axes, reference points, and measurements.**

| Aronyms/Abbreviation | Explanation |
| --- | --- |
| **Axes** | |
| x-axis | transverse axis (the outer product of the z and y axes) |
| y-axis | dorso-ventral axis (the normal vector of the z-axis, which runs between the midpoint of the tip of the first rib pairs) |
| z-axis | cranial-caudal axis (the vector between the first and fifth thoracic vertebrae) |
| **Reference points** | |
| ACJ | acromio-clavicular joint |
| glc | glenoid cavity (the midpoint between the supraglenoid tubercle and Infraglenoid tubercle.) |
| ifa | inferior angle |
| it | infraglenoid tubercle |
| mss | medial end of the scapula spine |
| SCJ | sterno-clavicular joint |
| spa | superior angle |
| st | supraglenoid tubercle |
| **Length measurements** | |
| BPD | biparietal diameter |
| CRL | crown-rump length |
| clavicle longitudinal length | length of segment ACJ-SCJ |
| scapula horizontal length | length of segment glc-mss |
| scapula vertical length | length of segment spa-ifa |
| **Angle measurements** | |
| ∠Clc | angle between the segment SCJ-ACJ and x-axis from the cranial view |
| ∠Clv | angle between the segment SCJ-ACJ and x-axis from the ventral view |
| ∠gll | angle between the segment st-it and y-axis from the lateral view |
| ∠Scc | angle between the segment mss-glc and x-axis from the cranial view |
| ∠Sc1c | angle between the segment spa-ifa and x-axis from the cranial view |
| ∠Sc1l | angle between the segment spa-ifa and y-axis from the lateral view |
| ∠Sc1v | angle between the segment spa-ifa and x-axis from the ventral view |
| ∠Sc2v | angle between the segment glc-ifa and x-axis from the ventral view |
| ∠T1Clc | angle between the segment ACJ-Th1 and x-axis from the cranial view |
| ∠T1Scc | angle between the segment glc-Th1 and x-axis from the cranial view |

## Length and angle measurements

The following lengths were measured: CRL, biparietal diameter (BPD), the length between the C4 and Th9 vertebrae, scapula horizontal length (segment glc-mss), scapula vertical length (segment spa-ifa), clavicle longitudinal length (segment ACJ-SCJ), and the width between bilateral ACJ, acromions, glc, ifa, mss, SCJ, and spa. Additionally, the following angles were also calculated: ∠Clv—the angle between the segment SCJ-ACJ and x-axis from the ventral view; ∠Sc1v - the angle between the segment spa-ifa and x-axis from the ventral view; ∠Sc2v - the angle between the segment glc-ifa and x-axis from the ventral view; ∠Sc1l - the angle between the segment spa-ifa and y-axis from the lateral view; ∠gll—the angle between the segment st-it and y-axis from the lateral view; ∠Clc—the angle between the segment SCJ-ACJ and x-axis from the cranial view; ∠Scc—the angle between the segment mss-glc and x-axis from

the cranial view; ∠T1Clc—the angle between the segment ACJ-Th1 and x-axis from the cranial view and ∠T1Scc—the angle between the segment glc-Th1 and x-axis from the cranial view.

## Results

### Morphogenesis of the scapula

**The blastemal and chondrogenous phases of the scapula at CS17 and CS20.** The scapula was detected in all three samples, but not in the 3-D reconstructed sample because the border was not sharp at CS17. The scapula body had high signal intensity at the contour and low signal intensity at the internal portion in PCX-CT images after CS18. The scapular body and coracoid as well as the humeral head were continuous and chondrified at CS18 (N = 3) (Fig 1A, CS18). The connection between the body and the coracoid process was vague in 2 out of the 3 samples (Fig 1B). The acromion was evident at CS19 (N = 3). The body was triangular in shape at CS20 (N = 3), which is similar to that observed in adults. The acromion was elongated and the acromio-clavicular joint was beginning to develop at this stage. The coracoid process was elongated and bent, which is similar to that observed in adults. The spine of the scapular was not formed until the end of the embryonic period [CS23 (N = 3)].

**Initial ossification at the scapula body (CS23 and early fetal period).** The scapula had low signal intensity in MRI images at the proximal part of the acromion connected to the scapular body at CS23 (N = 3). Initial ossification was detected at this part in early fetal samples with CRL 30 mm (Fig 2). Ossification was detected in 11 out of 13 samples with CRL 30–44 mm. This ossified region spread fanwise both laterally and medially during development.

**Growth of the scapula (fetal period).** The spine of the scapula was evident in samples with CRL ≥ 72 mm (N = 10) (black arrowhead, Fig 3). The spread of the fan-shaped ossification and the spine elongation were parallel to each other. The proximal part of the acromion was ossified in samples with CRL ≥ 100 mm (N = 9) (asterisk in Fig 3). The ossification reached the superior angle in samples with CRL 135 mm. The spine elongated up to the medial border in samples with CRL 200 mm. The glenoid, coracoid process, and medial border of the body were not ossified during this observation period (CRL ≤ 225 mm).

**Clavicle.** The clavicle had a high surface intensity signal in PCX-CT images from CS19 (N = 3), which indicated a membranous ossification (Fig 4). The elongated clavicle was located close to the sternum medially and to the acromion laterally at CS20 (N = 3). An extremely high signal intensity was observed in the center of the lateral two-third of the clavicle at CS20 (yellow arrowheads in Fig 4).

**Length measurements.** Both scapular horizontal and vertical lengths, clavicle longitudinal length, the length between the C4 and Th9 vertebrae, and the BPD increased linearly with CRL (strong correlation indicated by R>0.99) (Table 3). The width between each landmark on the scapula (acromion, glc, ifa, mss, and spa) and clavicle (ACJ and SCJ) also increased linearly with CRL (strong correlation indicted by $R^2$>0.97). Additionally, the BPD, an indicator of the brain width, and the width between the acromions, ACJ, and glc increased linearly with similar slopes (0.24 for the BPD and glc, 0.25 for acromions, and 0.23 for ACJ). The slopes between the BPD and the width between the acromions, ACJ, and glc were 0.96, 0.95, and 0.99, respectively, which indicate that the BPD and those three parameters increased in a similar manner and had similar values.

**3-D features of the shoulder girdle.** *3-D reconstruction.* 3-D reconstruction of the shoulder girdle (ventral and lateral views) in embryonic (CS19 (N = 3) and CS20 (N = 3)) and fetal (CRL 72 and 225 mm (N = 10)) periods revealed its positional features (Fig 5, S1–S4 Movies). For instance, the clavicle was located in a similar position irrespective of CS or CRL, while the

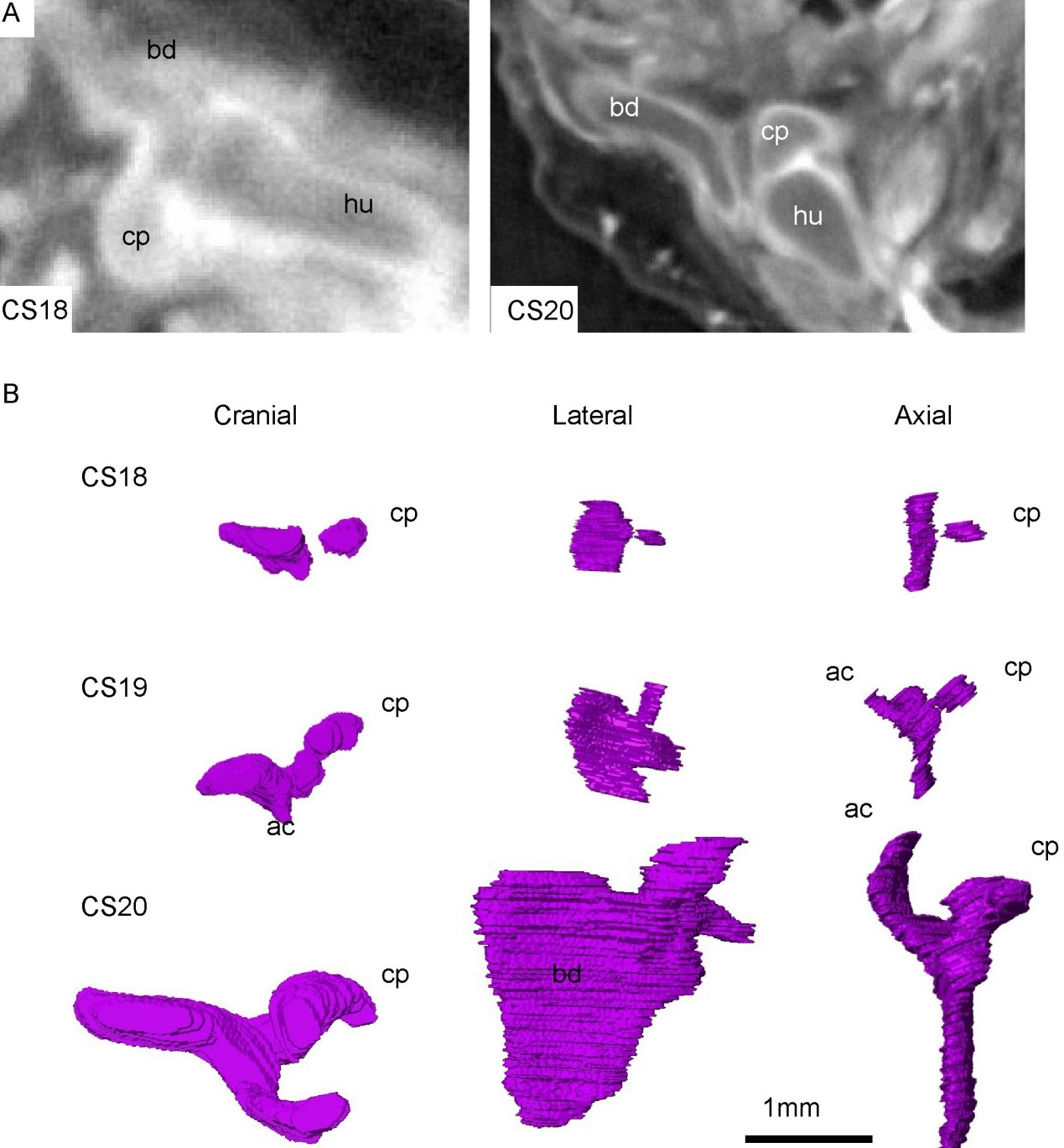

**Fig 1. Morphogenesis of the scapula at the blastemal and chondrogenous (during CS18 and CS20) phases.** A) Representative PCX-CT image of the scapula and humerus at CS18 and CS20. B) 3-D reconstruction of the scapula–cranial, lateral, and axial views. For each stage, three representative samples were analyzed. ac: Acromion, cp: Coracoid process, bd: scapula body, hu: humeral head.

scapula grew toward the caudal direction. Particularly, the spa and glc of the scapula were located at the cranial side of the x and y axes, while the scapula body was located at the caudal side of the x and y axes from the ventral and lateral views. The glenoid cavity was widely visible from the ventral view, which indicates that the cavity was ventrally oriented. The glenoid cavity

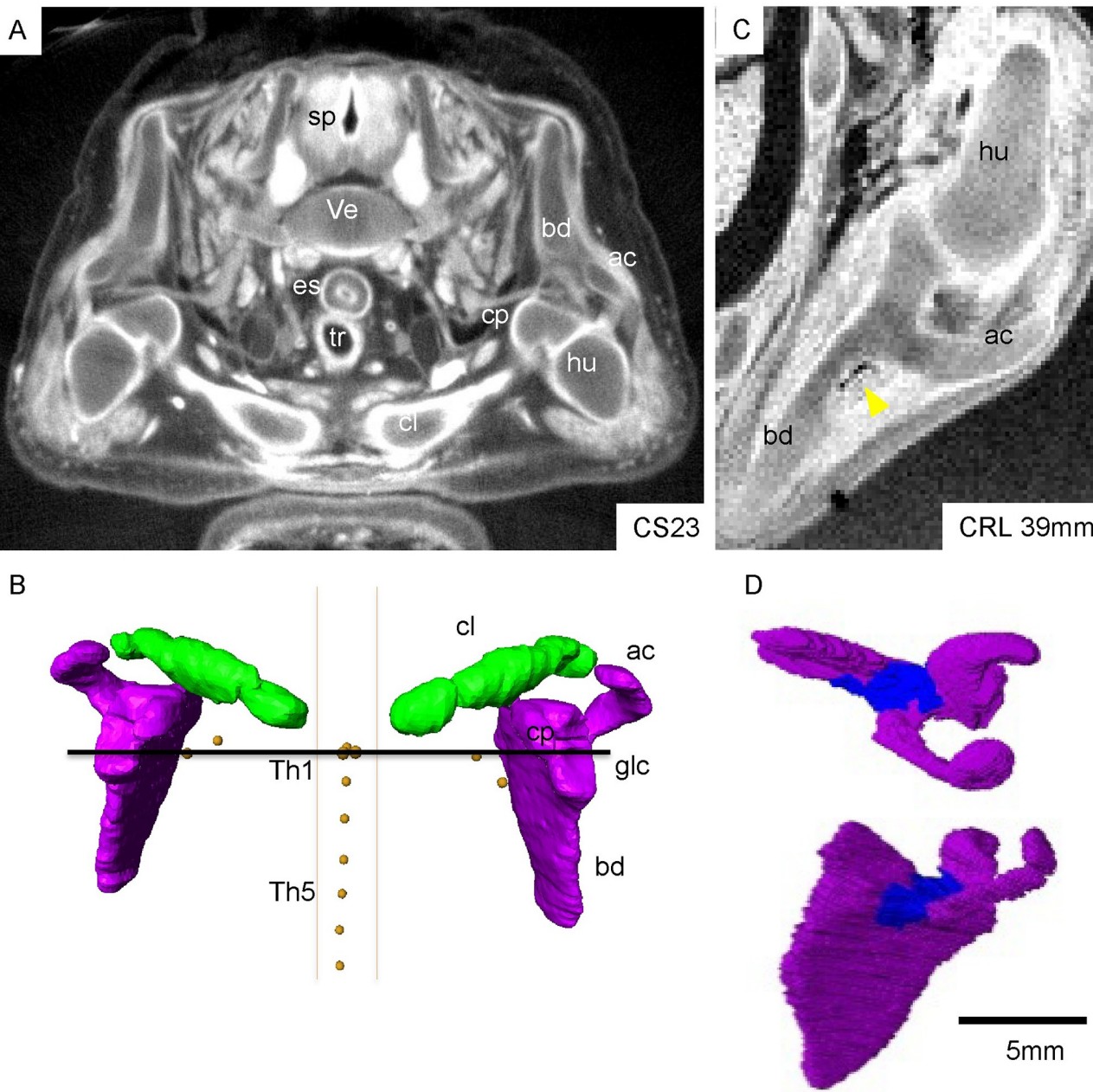

**Fig 2. Morphogenesis of the shoulder girdle at CS23 and early fetal period.** A) A representative transverse section of a PCX-CT image at CS23 showing the shoulder joint and clavicle. B) 3-D reconstruction of the scapula and clavicle at CS23 (ventral view). The horizontal line indicates the x-axis. Note that the clavicle was located at the cervical region (above the Th1 plane), and bilateral scapula bodies were almost parallel. The glenoid cavity was orientated anteriorly and slightly laterally in the cranial direction. Three samples at CS23 were observed. C) The scapula and shoulder joint in a fetal specimen with CRL 39 mm. Yellow arrowhead indicates the initial ossified region at the dorsal side of the body, close to the proximal part of the acromion. D) 3-D reconstruction of the scapula in a fetal specimen with CRL 39 mm. The ossified region is shown in blue. ac: Acromion, bd: scapula body, cl: clavicle (green), cp: Coracoid process, es: esophagus, glc; glenoid cavity, hu: humeral head, sp: spinal cord, Th1; first thoracic vertebra, Th5, fifth thoracic vertebra, tr; trachea, ve: vertebrae.

was oriented cranially from the lateral view. The medial border was oblique to the z-axis from both the ventral and lateral views. From the cranial view, both the clavicle and scapula were located on the ventral side of the x-axis in the embryonic period. Those positions changed during development, and both ACJ and glc were located close to the x-axis.

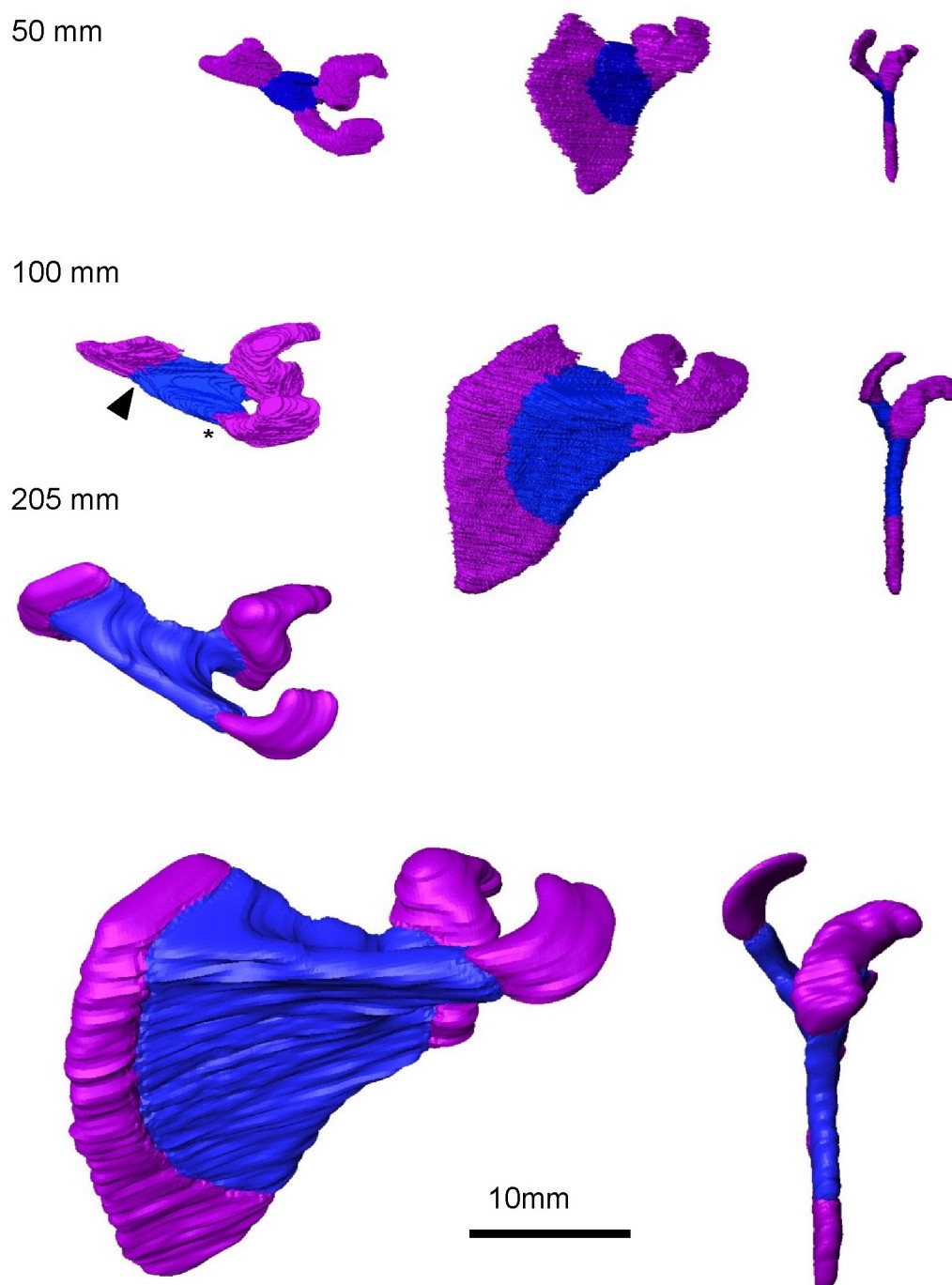

50 mm

100 mm

205 mm

**Fig 3. 3-D reconstruction of the scapula in the fetal period [CRL 50–205 mm (N = 12]).** Ossified regions are shown in blue. Black arrowhead and asterisk indicate the spine of the scapula and proximal part of the acromion, respectively.

*Position of the shoulder girdle along the z-axis (axial position).* The scapula was located above the level of the Th1 vertebra in the embryonic period (N = 23) (Fig 6). The approximate axial position of the spa and ifa was at the C6 and Th1 vertebrae, respectively, in the early embryonic period (N = 6). The location of the ifa changed from Th1 to Th5 in the late

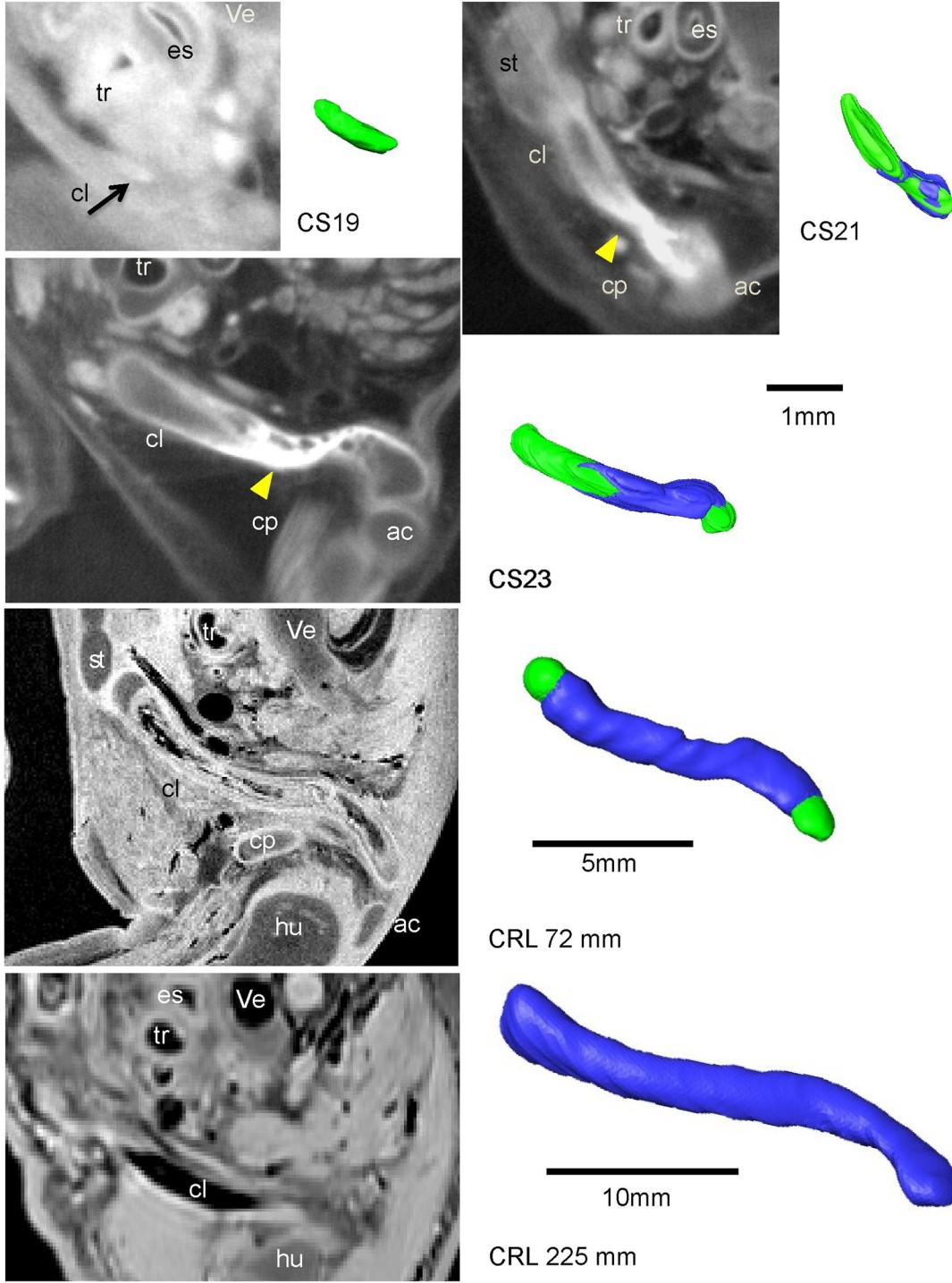

**Fig 4. PCX-CT image showing the clavicle at CS19 (N = 3) and CS21 (N = 2) and MR image at CS23 (N = 3) and of samples with CRL 72 mm and 225 mm (N = 10).** 3-D reconstructed clavicles are shown on the right side of each PCX-CT image. Blue indicates ossification. The black arrow indicates the initial ossification of the clavicle at CS19 (N = 3). Yellow arrowheads indicate ossification. ac: acromion, cl: clavicle (green), cp: Coracoid process, es; esophagus, hu: humeral head, st; sternum ve; vertebrae, tr; trachea.

**Table 3. Summary of length measurements in all specimens.**

| Length (mm) | range | Simple regression analysis | | | | | |
| --- | --- | --- | --- | --- | --- | --- | --- |
| | | between CRL | | | between BPD | | |
| | | slope (a) | y-intercept | R | slope (a) | y-intercept | R |
| Crown-Rump Length (CRL) | 14.1–225 | - | - | - | - | - | - |
| biparietal diameter (BPD) | 3.77–53.4 | 0.24 | 1.1 | 0.99 | - | - | - |
| C4 -Th9 Length | 3.82–50.8 | 0.23 | 0.97 | 0.99 | - | - | - |
| Scapula | | | | | | | |
| horizontal length (glc-mss) | 0.63–24.2 | 0.11 | -0.18 | 0.99 | - | - | - |
| vertical length (spa-ifa) | 0.96–30.7 | 0.14 | -0.34 | 0.99 | - | - | - |
| width between bilateral | | | | | | | |
| • acromions | 4.13–51.7 | 0.25 | 0.06 | 0.98 | 0.96 | -0.81 | 0.98 |
| • glenoid cavities (glc) | 3.50–49.6 | 0.24 | 0.6 | 0.99 | 0.99 | -0.26 | 0.99 |
| • superior angles (spa) | 3.03–28.4 | 0.12 | 1.6 | 0.99 | - | - | - |
| • inferior angles (ifa) | 3.78–43.7 | 0.2 | 1.87 | 0.99 | - | - | - |
| • medial end of the scapula spine (mss) | 2.95–30.9 | 0.13 | 1.84 | 0.98 | - | - | - |
| Clavicle | | | | | | | |
| longitudinal length | 0.62–28.1 | 0.13 | 0.47 | 0.99 | - | - | - |
| width between bilateral | | | | | | | |
| • AC joints(ACJ) | 2.53–47.9 | 0.23 | 0.67 | 0.99 | 0.95 | -0.17 | 0.98 |
| • SC joints (SCJ) | 0.60–9.00 | 0.04 | 0.44 | 0.97 | - | - | - |

C4, fourth cervial vertebrae; Th9, ninth thoracic vertebrae; AC, acromioclavicular; SC, Sterno-clavicular.

embryonic period (N = 17). Moreover, the location of the spa slightly descended to C7-Th1 while the location of the ifa descended to Th5-Th8 in the fetal period (N = 13). These data were consistent with the observation made during 3-D reconstruction from the ventral and lateral views that the scapula, instead of descending, grew toward the caudal direction.

The axial position of three joints, glc, SCJ, and ACJ was mostly constant except in the embryonic period. The glc was located around the C7 level. The SCJ was located at the C5-C7 level in the embryonic period, and this location slightly descended to the C7 level in the fetal period. The ACJ was located at the C5-C6 level in the embryonic period, and this location remained constant in the fetal period.

*Angles on the x and y axes (transverse plane) from the ventral and lateral views*. The angle of the clavicle segment ACJ-SCJ on the x-axis from the ventral view (∠Clv) was constantly around 15 degrees, and varied in the embryonic period (between -6.0 and 20.0 degrees) (N = 23) (Fig 7). This angle was occasionally high in large samples (between 25.5 and 37.7 degrees)(N = 3). The angle of the segment spa-ifa on the x-axis from the ventral view (∠Sc1v) was constantly around 70 degrees in this study. The angle of the segment glc-ifa on the x-axis from the ventral view (∠Sc2v) was between 70 and 80 degrees in the embryonic period (N = 23). This angle slightly increased and was constantly around 85 degrees in the fetal period (N = 13).

The angle of the segment spa-ifa on the y-axis from the lateral view (∠Sc1l) was between 100 and 110 degrees in the early embryonic period (N = 5), and decreased to (90 and 100 degrees) in the late embryonic period (N = 18). This angle was between 65 and 80 degrees in the fetal period (N = 13). The angle of the segment st-it on the y-axis from the lateral view (∠gll) was between 65 and 85 degrees in the embryonic period, and the angle decreased (between 50 and 65 degrees) in the fetal period.

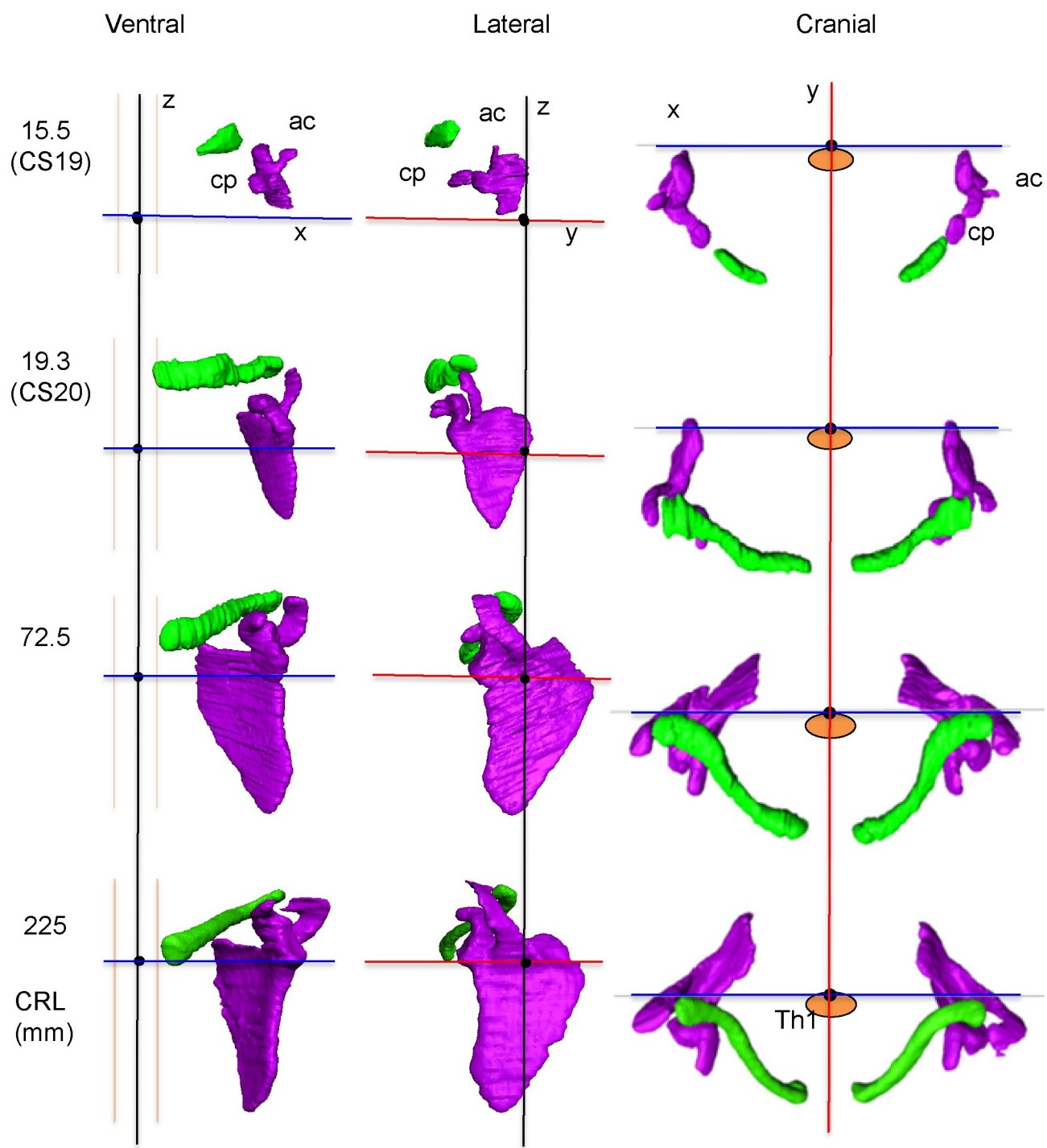

**Fig 5. Ventral, lateral, and cranial views of the 3-D reconstruction of the scapula (purple) and clavicle (green) in the embryonic period (CS19 (N = 3) and CS20 (N = 3)) and fetal period (CRL 72.5 and 225 mm (N = 10)).** Blue line; x-axis, red line; y-axis, black line; z-axis. Th1: first thoracic vertebra, ac: acromion, cp: Coracoid process.

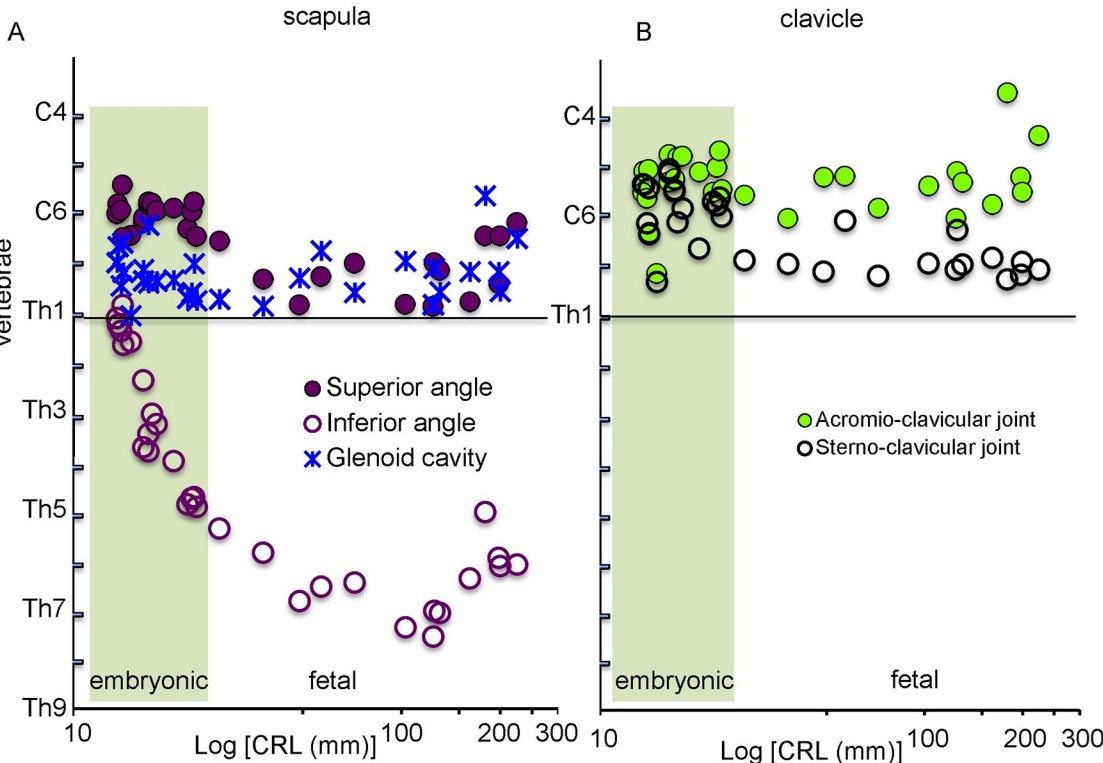

**Fig 6. Position of the scapula (A) and the clavicle (B) along the z-axis.** Solid purple circles indicate the superior angle of the scapula (spa), open purple circles indicate the inferior angle of the scapula (ifa), blue stars indicate the center of the glenoid cavity (glc), solid green circles indicate the streno-clavicular joint (SCJ), open black circles indicate the acromio-clavicular joint (ACJ). The position of the glenoid cavity was calculated as the midpoint between the Infraglenoid and supraglenoid tubercle. Embryonic period (N = 23), Fetal period (N = 13).

*Angles on the x-axis from the cranial view*. The angle of the clavicle (segment ACJ-SCJ) on the x-axis from the cranial view (∠Clc) was constantly between 30 and 45 degrees, and varied in the embryonic period (15.5 to 46.1 degrees) (N = 23) (Fig 8A). The angle of the segment glc-mss on the x-axis from the cranial view (∠Scc) varied in this study; between 60 and 75 degrees in the early embryonic period (N = 5), increased to between 75 and 100 degrees in the late embryonic period (N = 18), and decreased to between 45 and 75 degrees in the fetal period (N = 13).

*Position of the shoulder girdle to the body trunk (vertebrae)*. Both angles of the segments ACJ-Th1 (∠T1Clc) and glc-Th1 (∠T1Scc) on the x-axis from the cranial view decreased with development (Fig 8B). The ∠T1Clc was between 30 and 45 degrees in the embryonic period (N = 23), and gradually decreased (between -5 and 15 degrees) in large CRL fetal samples (N = 10). The ∠T1Scc was between 20 and 40 degrees in the embryonic period, and gradually decreased (between 5 and 20 degrees) in large CRL fetal samples (N = 10). These data indicate that both ACJ and glc changed from the anterior-lateral to the lateral side with development.

*Relationship between the scapula and the clavicle*. During initial morphogenesis, the angles between the scapula and clavicle were obtuse from the ventral (∠Sc1v+∠Clv) and cranial (∠Sc1c+∠Clc) views (Fig 9). These angles reduced with development from embryonic to fetal period. The angle from the cranial view (∠Sc1c+∠Clc) was almost rectangular, while that from the ventral view (∠Sc1v+∠Clv) was acute. In contrast, the angle between the segments ACJ-Th1 and glc-Th1 remained almost the same (between 0 and 5 degrees) in the embryonic period (N = 23) and slightly increased with development (between 5 and 15 degrees)(N = 13).

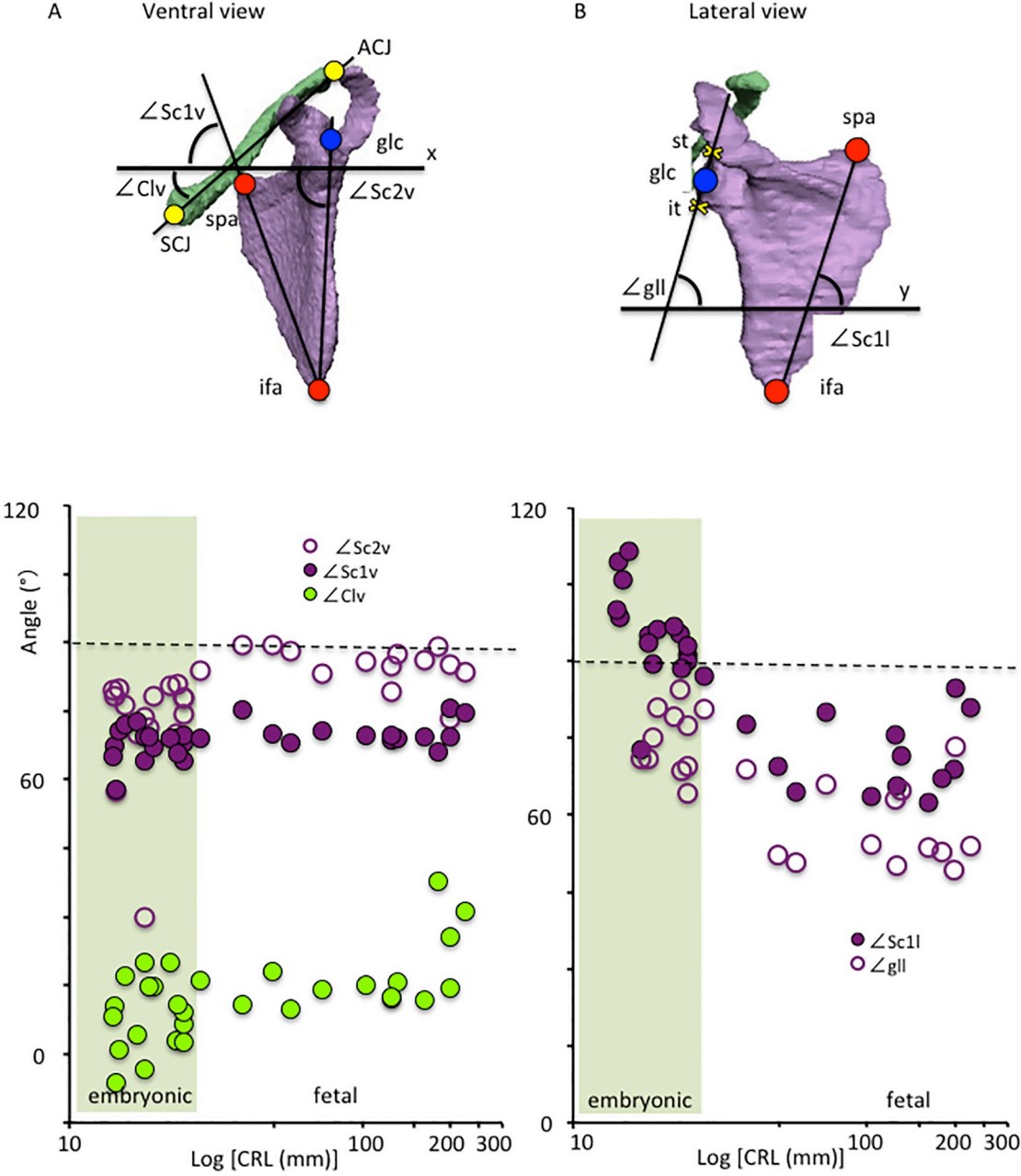

**Fig 7. Angle measurements of shoulder girdle from the ventral view (A) and lateral view (B).** A) The angle of the scapula and clavicle on the x-axis from the ventral view (xz plane) during development. Angles of the segments SCJ-ACJ (∠Clv) (solid green circle), spa-ifa (∠Sc1v) (solid purple circle), and glc-ifa (∠Sc2v)(open purple circle) on the x-axis from the ventral view were measured. B) The angle of the scapula and clavicle on the y-axis from the lateral view during development. Angles of the segment spa-ifa (∠Sc1l) (solid purple circle) and st-it (∠gll) (open purple circle) on the y-axis from the lateral view were measured. Embryonic period (N = 23), Fetal period (N = 13). ACJ: Acromio-clavicular joint, glc: glenoid cavity (midpoint between st and it), ifa: Inferior angle, it: Infraglenoid tubercle, SC: Sterno-clavicular joint, spa: Superior angle, st: supraglenoid tubercle.

## Discussion

Lewis (1902) described the development of the upper limb including the shoulder girdle at the beginning of the last century [5]. Reconstruction of the skeleton, muscle masses, and nerves at

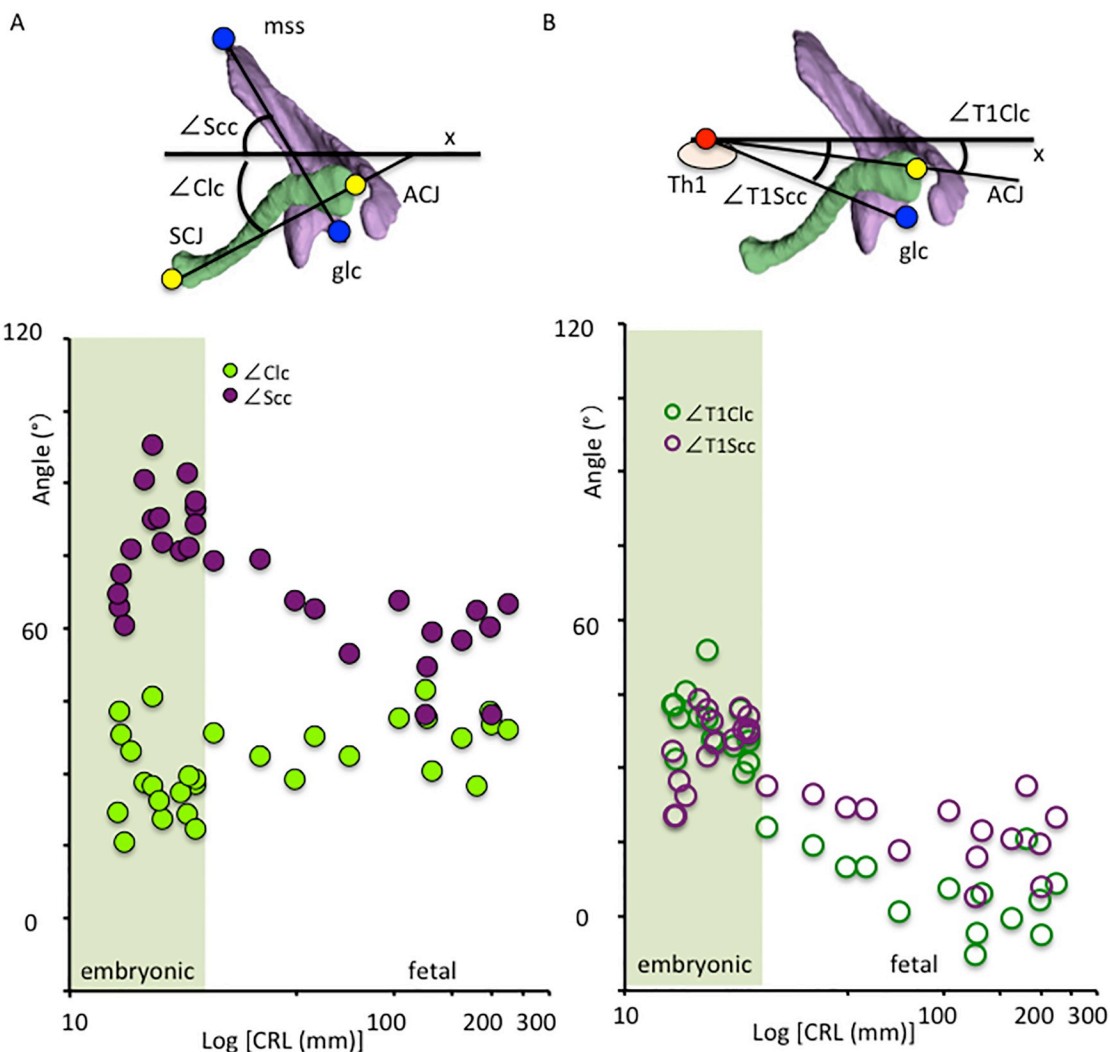

**Fig 8. Angle measurements of the shoulder girdle from the cranial view.** A) The angle of the scapula and clavicle on the x-axis from the cranial view during development. Solid green circles indicate the angle of the segment SCJ-ACJ on the x-axis from the cranial view (∠Clc), solid purple circles indicate the angle of the segment mss-glc on the x-axis from the cranial view (∠Scc). B) Relationship between the shoulder girdle and the body trunk from the cranial view. Open purple circles indicate the angle of the segment ACJ-Th1 (∠T1Clc) on the x-axis from the cranial view, while open green circles indicate the angle of the segment glc-Th1 (∠T1Scc) on the x-axis from the cranial view. Embryonic period (N = 23), Fetal period (N = 13). ACJ: Acromio-clavicular joint, glc: glenoid cavity (midpoint between st and it), mss: medial end of the scapula spine, SCJ: Sterno-clavicular joint, Th1: first thoracic vertebra.

CS16, CS18, CS19, and CS22 was illustrated by the author; however, each skeletal form including the scapula lacked close-up illustrations, and they were shown as a part of the whole upper limb system in each figure. Additionally, the initial detection of the scapula in recent studies had been performed at one or two stages later than that described in Lewis's study. Mekonen et al. detected the scapula (shoulder blade) at CS17 in histological sections [29]. Hita-Contreras et al. observed the chondrogenic anlagen at the medial border of the scapula from as early as CS17, whereas the scapula body appeared at CS18 [10]. We first detected the scapula body at CS17 with the coracoid and humeral head at CS18 on PCX-CT images, consistent with recent histological studies. The detection of the initial blastemal and chondrogenous phases of the

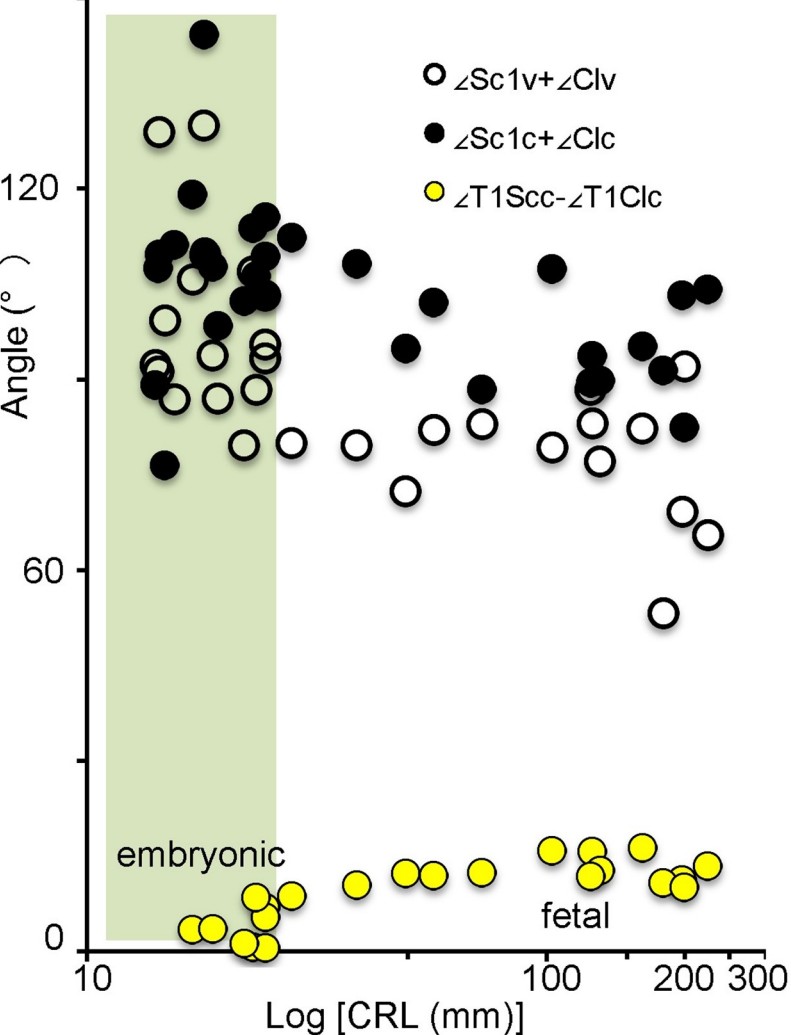

**Fig 9. Relationship between the scapula and the clavicle from the cranial view.** Open black and solid black circles indicate angles between the scapula and clavicle from the ventral (∠Sc1v+∠Clv) and cranial (∠Sc1c+∠Clc) views, respectively. Solid yellow circles indicate angles between the segment glc-Th1 and ACJ-Th1 from the cranial view (∠T1Scc-∠T1Clc). Embryonic period (N = 23), Fetal period (N = 13).

bones using the PCX-CT was comparable with that of other locomotive structures such as the pelvis and femur [30, 31].

In this study, the connection of the body and coracoid could not be confirmed at CS18 because of the resolution of the PCX-CT image. Hita-Contreras et al. observed three outgrowths of the mesenchymal condensation with an irregular shape corresponding to the body of the scapula, future coracoid process, and large mass of the humerus, acromion, and spine of the scapula [10]. The authors, however, did not mention whether these three outgrowths were connected. Lewis (1902) illustrated the coracoid without the scapula body at CS18, and showed that they are connected at CS19 [5]. The morphogenesis of the coracoid is different from the scapula body in many aspects. First, the ossification of the coracoid begins after birth, which is different from the scapula body. Between 15th and 18th months after birth, ossification occurs in the middle of the coracoid process, which joins with the rest of the bone at 15 years of age [32, 33]. Recent studies with genetic approach indicated that the development of different

parts of the scapula such as the coracoid, acromion, and scapula body are controlled by different genes, and that these parts of the scapula may derive from different anlagens [13]. The coracoid is separated from scapula in most of the vertebrates [34]. It may be interesting to elucidate whether the scapula body and coracoid are separate in the initial blastemal and chondrogenous phases at CS18.

Lewis (1902) paid particular attention to the axial position of the scapula in his study [5], and showed that the scapula was located at the level of 4[th] cervical intercostal space at CS16, between the C4 and Th1 vertebrae at CS18, and largely caudal to Th1 vertebrae with the inferior angle extending to the Th5 vertebra at CS19. A very small portion of the scapula is cranially positioned to the level of the rib 1 and the inferior angle is at the level of 5th intercostal space at CS21. Several other studies also described similar characteristics of the axial position of the scapula [6, 7]. The elevation of the scapulae can be of clinical interest because a persistent elevation of the scapula is commonly associated with Sprengel's deformity [8].

Müller and O'Rahilly (1986) [7] indicated that the scapula is enlarged in the embryonic period, although it does not descend. Our 3-D reconstruction and morphometric data clearly support such indications. The landmarks on scapula and clavicle (Superior angle, Sterno-clavicular joint, Acromio-clavicular joint, and Glenoid cavity) remained at a similar axial position while only the axial position of the Inferior angle reduced, which indicates that the scapula enlarged in the caudal direction but did not descend. The axial position of the Inferior angle is at the level of the Th7 vertebra while that of the scapula spine is at the level of the Th3 vertebra in adults. Our analyses indicate that the scapula may not descend even after the fetal period, which is contrary to previously published results [7]. Our finding that the scapula enlarged, but did not descend, implies that the congenital elevation of the scapula is not due to the failure of scapular "descent" during development [7, 14].

The 3-D reconstruction and morphometry in our study revealed a unique position, other than the axial position, of the shoulder girdle in the embryonic and fetal periods. In contrast to the constant position of the clavicle, the scapula body rotated internally and upward at the initiation of morphogenesis. Thus, the right and left scapula bodies seemed to be almost parallel to each other, which is consistent with the findings observed in Fig 2 of O'Rahilly et al. [8]. The internal rotation of the scapula changed externally while the upward rotation remained unchanged. In comparison to adults, the scapula rotated internally and upward in the fetal period. The shoulder girdle was located at the ventral side of the body trunk (vertebrae) at the time of initial morphogenesis, and subsequently, changed position to the lateral side of the vertebrae in the late embryonic and fetal periods. Shoulder girdle position observed in the fetal period can be consistent with that in adults. Such a unique position of the shoulder girdle may contribute to the stage-specific posture of the upper limb, which is one of the important external features in determining staging, especially between CS18 and CS23 [24]. For example, the axial skeleton (vertebrae) becomes straight and the upper limb extends vertically to the axial skeleton at CS19. The shoulder (humeral head) becomes evident externally, and the joint flexed with the elbow becomes pronated at CS23. The posture at this stage can be explained, in part, by the unique positional changes of the scapula in the embryonic period.

The present study indicates that the biparietal diameter and the width between acromions, acromio-clavicular joints, and glenoid cavity, which are indicators of the shoulder girdle width, grew in parallel and had similar values during development. Adequate internal and upward rotation of the scapula could reduce the shoulder width, thereby facilitating childbirth. Considering that the abnormally large shoulder-to-head ratio can result in shoulder dystocia and clavicle fracture [18, 19], the data presented in this study can be used as normal morphometric references for shoulder girdle evaluation in embryonic and fetal periods.

This study has a few limitations. First, the border of the shoulder girdle in the blastemal and chondrogenous phases was obscure in digitalized data obtained with the PCX-CT system [30, 31], which made it difficult to segment and determine the connection. For example, the connection between the scapula body and coracoid was not determined at CS18. Second, the z-axis was defined using the Th1 and Th5 vertebrae. Changes in the cervical and thoracic flexion were observed in the embryonic period [24], which could affect the orientation of the z-axis. Third, the samples used in this study were fixed using a medium that contains formaldehyde and stored in the sample tube for a long period. The posture of the sample, especially the limbs, are prone to alterations due to the application of fixatives and external force. Finally, although our samples were classified as normal based on external morphology, there is no guarantee that all samples had normal development.

In conclusion, this study provides useful 3-D positional morphometry of the shoulder girdle, which can aid in a comprehensive understanding of the shoulder girdle development and differentiating between normal and abnormal developments.

## Supporting information

**S1 Movie. 3-D reconstruction of the shoulder girdle at CS19.**
(MPG)

**S2 Movie. 3-D reconstruction of the shoulder girdle at CS20.**
(MPG)

**S3 Movie. 3-D reconstruction of the shoulder girdle at CS23.**
(MPG)

**S4 Movie. 3-D reconstruction of the shoulder girdle in the fetal period (CRL, 225 mm).**
(MPG)

**S1 File.**
(XLSX)

## Acknowledgments

The authors thank Ms. Chigako Uwabe and Dr. Haruyuki Makishima at the Congenital Anomaly Research Center for their technical assistance in handling the human embryos. This work was performed with approval from the Photon Factory Program Advisory Committee (Proposal No. 2019G542, 2017G541).

## Author Contributions

**Conceptualization:** Tetsuya Takakuwa.

**Data curation:** Sayaka Tanaka, Rino Sakamoto.

**Formal analysis:** Sayaka Tanaka, Rino Sakamoto.

**Funding acquisition:** Tetsuya Takakuwa.

**Project administration:** Tetsuya Takakuwa.

**Resources:** Shigehito Yamada, Hirohiko Imai, Akio Yoneyama.

**Visualization:** Toru Kanahashi.

**Writing – original draft:** Tetsuya Takakuwa.

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
