## [Decision Letter · Decision Letter 0]

22 Jul 2020

PONE-D-20-11726

The shoulder girdle formation and position in the embryonic and early fetal periods

PLOS ONE

Dear Dr. Takakuwa,

Thank you for submitting your manuscript to PLOS ONE. After careful consideration, we feel that it has merit but does not fully meet PLOS ONE’s publication criteria as it currently stands. Therefore, we invite you to submit a revised version of the manuscript that addresses the points raised during the review process.

As you can see, both reviewers appreciated the significance and scientific rigor of the manuscript, but they also suggested to have the manuscript edited for its English language, preferably by a native speaker or professional editing service. Furthermore, as the manuscript deals with human embryos, more sensitive language should be used to avoid potential controversies and negative impact to the journal and the authors. For instance, the text in lines 110-118 is strong and could be an issue for the journal. The specimens used are scientific and seem to be ethically collected, but the statement that "written consent was not obtained from all parents" needs to be clarified. If verbal consent was obtained and recorded from all parents with the approval of the IRB, then this statement may be deleted. Words such as "abortion" or "aborted fetuses" should be avoided.

You should provide the approval number (from your ethics committee) to validate/demonstrate proper supervision of your study.

We look forward to receiving your revised manuscript.

Kind regards,

Nancy Beam, PhD

Senior Staff Editor

PLOS ONE

On behalf of:

Jianbo Wang, Ph.D.

Academic Editor

PLOS ONE

Journal Requirements:

2. Thank you for including your ethics statement:  "The ethics committee of the Kyoto University Faculty and Graduate School of Medicine approved this study, which used human embryo and fetal specimens (E986, R0316). Specimens stored at the Kyoto Collection were used in the study, which were acquired when pregnancy was terminated for socioeconomic reasons under the Maternity Protection Law of Japan. Parents provided their verbal informed consent to have these specimens deposited in the collection, and each participant’s consent was documented in the medical record. Written consent was not obtained from all parents. Samples were collected between 1963 and 1995 in accordance with relevant regulations of those time periods."

a.) Please amend your current ethics statement to address the following points:

     a) Please explain why was written consent was not obtained

     b) Did the ethics committees/IRBs approve this consent procedure?

     c) You state "Written consent was not obtained from all parents." Did they provide verbal consent as described?

     d) Were the samples anonymized and/or de-identified?

b.) Once you have amended this/these statement(s) in the Methods section of the manuscript, please add the same text to the “Ethics Statement” field of the submission form (via “Edit Submission”).

For additional information about PLOS ONE ethical requirements for human subjects research, please refer to http://www.PLOSone.org/static/guidelines.action#human.

3. We note you have included a table to which you do not refer in the text of your manuscript. Please ensure that you refer to Table 1 in your text; if accepted, production will need this reference to link the reader to the Table.

Reviewers' comments:

Reviewer's Responses to Questions

**Comments to the Author**

1. Is the manuscript technically sound, and do the data support the conclusions?

Reviewer #1: Partly

Reviewer #2: Yes

2. Has the statistical analysis been performed appropriately and rigorously? 

Reviewer #1: N/A

Reviewer #2: I Don't Know

3. Have the authors made all data underlying the findings in their manuscript fully available?

Reviewer #1: Yes

Reviewer #2: Yes

4. Is the manuscript presented in an intelligible fashion and written in standard English?

Reviewer #1: No

Reviewer #2: Yes

5. Review Comments to the Author

Reviewer #1: Dear Authors,

The Manuscript entitled "The shoulder girdle formation and position in the embryonic and early fetal periods" by Tanaka et al., serves to reconstruct the scapula and clavicle from imaging techniques performed on human embryonic samples in order to shed light on both bony elements' morphogenesis and positional development during embryogenesis. They have provided a nice detailed anatomical study on these two structures more often than not ignored by scientists studying the development of the appendages. Their studies are important contributions to understanding girdle formation, particularly in human samples, of which we know very little. The hope is that the same methods can be used to examine other skeletal structures of the axial and appendicular systems. With the above comments in mind, this Reviewer does have minor comments that need addressing. I have provided them line by line (according to line items in the manuscript document), but please note, some items should be corrected throughout the text. They are as follows:

Line 22-3: unclear what is meant by this sentence

Line 32: by "height", do you mean axial position? Fix throughout manuscript.

Line 60: please define better "Th2-Th7 height"

Lines 66-70: can this sentence be written more concisely and without direct reference in the text to the figures by Lewis?

Line 70: What is meant by "higher" - more cranial? Fix throughout by using cranial or caudal to indicate direction of shift.

Line 110: please reference a table and provide one.

Line 170: please correct by writing "ventral tip"

Lines 184-198: Can these measurements be made into a descriptive table?

Line 22: What is meant by "lowered" - Please fix throughout the text.

Line 243 (and others). It is best to have the authors provide the number of biological replicates next to each observation

they make (e.g., N=2). This is an important point, which should also be provided in each figure legend.

Line 288-290: Can this be defined better?

Line 304: Instead of "was around" use "was at the approximate positional height of the C6 and TH1 verebrae". Please correct instances where the language use is too casual. Be more scientifically accurate. Another example is Line 336: Remove "around" and say "between" but there are others as well.

Line 398: Say "Lewis (year)" - no need for WH.

Line 401: Say "each skeletal form"

Line 407: What is meant by "main part of the scapula"?

Line 433: The general scientific audience will not recognize what th1 and ifa are, and so it is best to use their full names and what they refer to here. Please correct all instances in the Discussion. It is okay for these descriptors to be used in the Results section provided they are defined at the first use. This will help the reader follow the text.

Line 465: write "humeral head"

Reviewer #2: In this study the authors have analyzed in greater depth the development of the shoulder girdle during embryonic and early fetal human development. The authors made use of 35 previously collected formaldehyde-fixed embryos/fetuses from the Kyoto University collection, and they visualized and analyzed the developing skeleton by a variety of imaging procedures including PCX-CT and MR. Serial 2D images were reconstructed into highly-detailed 3D images, and the elements of the shoulder girdle -scapula and clavicle- were developmentally monitored against anatomical landmarks, including vertebrae. The data suggest that the apical portion of the shoulder girdle primordium remained at the same height implying that the scapula enlarged caudally, reached its adult position in fetal life and rotated internally and upward. The girdle was initially located on the ventral side of the vertebrae, but changed its position to the lateral side in late embryonic and fetal periods. These and other data are proposed to serve as a normal morphometric reference for future evaluation of shoulder girdle development in normal and pathological circumstances.

The study is of significance since there is still quite limited understanding of shoulder girdle development in humans. This is particularly so with regard to morphogenesis of the girdle elements and their positioning and repositioning over embryonic and fetal development. Given the body of data and imaging reconstruction, it appears that the study was carried out with keen attention and great care.

Concerns are very minor.

The entire manuscript should be edited by a native speaker since it contains several grammatical and syntactical errors.

The authors use a large number of acronyms, particularly with regards to various anatomical angles. It would be helpful to include a Table listing all the acronyms.

A better title of the manuscript could be: “Shoulder girdle formation and positioning during embryonic and early fetal human development”.

6. PLOS authors have the option to publish the peer review history of their article (what does this mean?). If published, this will include your full peer review and any attached files.

Reviewer #1: No

Reviewer #2: No

---

## [Author Response · Author response to Decision Letter 0]

4 Aug 2020

PONE-D-20-11726

The shoulder girdle formation and position in the embryonic and early fetal periods

Reply to the editor

1. they also suggested to have the manuscript edited for its English language, preferably by a native speaker or professional editing service. 

The manuscript was edited for its English language, by professional editing service, Editage Corp.

2. As the manuscript deals with human embryos, more sensitive language should be used to avoid potential controversies and negative impact to the journal and the authors. 

1) For instance, the text in lines 110-118 is strong and could be an issue for the journal. The specimens used are scientific and seem to be ethically collected, but the statement that "written consent was not obtained from all parents" needs to be clarified. If verbal consent was obtained and recorded from all parents with the approval of the IRB, then this statement may be deleted. 

Because verbal consent was obtained and recorded from all parents with the approval of the IRB, the phrase "written consent was not obtained from all parents" was removed.

2) Words such as "abortion" or "aborted fetuses" should be avoided.

Words "abortion" or "aborted fetuses" was rephrased.

3) You should provide the approval number (from your ethics committee) to validate/demonstrate proper supervision of your study.

The approval number from your ethics committee to demonstrate proper supervision of our study was provided (E986, R0316).

2. Please amend your current ethics statement to address the following points:

a) Please explain why was written consent was not obtained

b) Did the ethics committees/IRBs approve this consent procedure? 

c) You state "Written consent was not obtained from all parents." Did they provide verbal consent as described? 

d) Were the samples anonymized and/or de-identified?

e)) Once you have amended this/these statement(s) in the Methods section of the manuscript, please add the same text to the “Ethics Statement” field of the submission form (via “Edit Submission”).

We amend our current ethics statement as indicated, in the Methods section of the manuscript, and added the same text to the “Ethics Statement” field of the submission form (via “Edit Submission”).

3. We note you have included a table to which you do not refer in the text of your manuscript. Please ensure that you refer to Table 1 in your text; if accepted, production will need this reference to link the reader to the Table.

Table 1 (Table 3 in the revised manuscript) was referred in the text of our manuscript.

Reply to Reviewer #1

1. Line 22-3: unclear what is meant by this sentence

The sentence was amended as follows;

However, such data are limited except for information on the axial position of the scapula.

2. Line 32: by "height", do you mean axial position? Fix throughout manuscript.

Height was reworded using the term “axial position” through out the manuscripts.

3. Line 60: please define better "Th2-Th7 height"

The phrase "Th2-Th7 height" was rephrased as “the axial position between the Th2 and Th7 vertebrae”.

4. Lines 66-70: can this sentence be written more concisely and without direct reference in the text to the figures by Lewis?

This sentence was written more concisely as follows;

Lewis (1902) [5] precisely described the morphogenesis of the upper limb in the embryonic period, including the skeleton, muscles, and nerves. The scapula and clavicle were illustrated as skeletal components.

5. Line 70: What is meant by "higher" - more cranial? Fix throughout by using cranial or caudal to indicate direction of shift.

The word “higher” was removed and the sentence was changed as follows;

Regarding the scapula, the author paid particular attention to the cranial position of the scapula in the initial blastemal and chondrogenous phases [5].

6. Line 110: please reference a table and provide one.

Table 1 was provided.

7. Line 170: please correct by writing "ventral tip"

The word “tip” was rephrased as “ventral tip” as indicated.

8. Lines 184-198: Can these measurements be made into a descriptive table?

Table 2 was provided.

9. Line 222: What is meant by "lowered" - Please fix throughout the text.

The sentence was changed as follows;

The scapula had low signal intensity in MRI images at the proximal part of the acromion connected to the scapular body at CS23 (N=3).

10. Line 243 (and others). It is best to have the authors provide the number of biological replicates next to each observation they make (e.g., N=2). This is an important point, which should also be provided in each figure legend.

The number of biological replicates next to each observation was provided in texts and each figure legend.

11.Line 288-290: Can this be defined better?

The sentence was changed as follows;

Particularly, the spa and glc of the scapula were located at the cranial side of the x and y axes, while the scapula body was located at the caudal side of the x and y axes from the ventral and lateral views.

12. Line 304: Instead of "was around" use "was at the approximate positional height of the C6 and TH1 verebrae". Please correct instances where the language use is too casual. Be more scientifically accurate. 

The sentence was changed as follows;

The approximate axial position of the spa and ifa was at the C6 and Th1 vertebrae, respectively, in the early embryonic period (N=6).

The word “around” was reworded through out the texts.

13. Line 336: Remove "around" and say "between" but there are others as well.

The sentence was changed as indicated.

The angle of the segment spa-ifa on the y-axis from the lateral view (∠Sc1l) was between 100 and 110 degrees in the early embryonic period (N=5), and decreased to (90 and 100 degrees) in the late embryonic period (N=18).

14. Line 398: Say "Lewis (year)" - no need for WH.

The phrase was changed as indicated.

Line 401: Say "each skeletal form"

The phrase was changed as indicated.

Line 407: What is meant by "main part of the scapula"?

The phrase was rephrased as “the scapula body”.

Line 433: The general scientific audience will not recognize what th1 and ifa are, and so it is best to use their full names and what they refer to here. Please correct all instances in the Discussion. It is okay for these descriptors to be used in the Results section provided they are defined at the first use. This will help the reader follow the text.

Most of abbreviations used in Discussion were spell out

Line 465: write "humeral head"

The phrase was changed as indicated.

Reply to Reviewer #2: 

1. The entire manuscript should be edited by a native speaker since it contains several grammatical and syntactical errors.

The manuscripts were subjected to the English language editing service (Editage).

2. The authors use a large number of acronyms, particularly with regards to various anatomical angles. It would be helpful to include a Table listing all the acronyms.

Table 2 listing all the acronyms were provided. 

3. A better title of the manuscript could be: “Shoulder girdle formation and positioning during embryonic and early fetal human development”.

The title was changed as indicated.

---

## [Editor Report · Decision Letter 1]

13 Aug 2020

Shoulder girdle formation and positioning during embryonic and early fetal human development

PONE-D-20-11726R1

Dear Dr. Takakuwa,

We’re pleased to inform you that your manuscript has been judged scientifically suitable for publication and will be formally accepted for publication once it meets all outstanding technical requirements and correct the following grammatical errors:

1.     Please revised the following sentence in the “Ethics Statement” as it is not comprehensible: “and each participantocioeconomic reasons and in accordance with the MAll samples were anonymized and de-identified.”

            2.     The following sentence is duplicated on page 5; line 64-65:

                  “The medial border is almost parallel to the vertebrae column (the cranial-caudal axis). 

            3. Page 9, line 123-124: “and each participantocioeconomic reasons and in accordance with    the MAll samples were anonymized and de-identified.”

Kind regards,

Jianbo Wang, Ph.D.

Academic Editor

PLOS ONE
---

## [Editor Report · Acceptance letter]

27 Aug 2020

PONE-D-20-11726R1 

Shoulder girdle formation and positioning during embryonic and early fetal human development 

Dear Dr. Takakuwa:

I'm pleased to inform you that your manuscript has been deemed suitable for publication in PLOS ONE. Congratulations! Your manuscript is now with our production department. 

Kind regards, 

on behalf of

Dr. Jianbo Wang 

Academic Editor

PLOS ONE